# Emerging Quantitative Biochemical, Structural, and Biophysical Methods for Studying Ribosome and Protein–RNA Complex Assembly

**DOI:** 10.3390/biom13050866

**Published:** 2023-05-19

**Authors:** Kavan Gor, Olivier Duss

**Affiliations:** 1Structural and Computational Biology Unit, European Molecular Biology Laboratory (EMBL), 69117 Heidelberg, Germany; kavan.gor@embl.de; 2Faculty of Biosciences, Collaboration for Joint PhD Degree between EMBL and Heidelberg University, 69117 Heidelberg, Germany

**Keywords:** RNP assembly, ribosome assembly, protein–RNA interactions, RNA folding, assembly intermediates, in vitro reconstitutions, mass spectrometry, single-molecule fluorescence microscopy, cryo–electron microscopy, RNA structure probing

## Abstract

Ribosome assembly is one of the most fundamental processes of gene expression and has served as a playground for investigating the molecular mechanisms of how protein–RNA complexes (RNPs) assemble. A bacterial ribosome is composed of around 50 ribosomal proteins, several of which are co-transcriptionally assembled on a ~4500-nucleotide-long pre-rRNA transcript that is further processed and modified during transcription, the entire process taking around 2 min in vivo and being assisted by dozens of assembly factors. How this complex molecular process works so efficiently to produce an active ribosome has been investigated over decades, resulting in the development of a plethora of novel approaches that can also be used to study the assembly of other RNPs in prokaryotes and eukaryotes. Here, we review biochemical, structural, and biophysical methods that have been developed and integrated to provide a detailed and quantitative understanding of the complex and intricate molecular process of bacterial ribosome assembly. We also discuss emerging, cutting-edge approaches that could be used in the future to study how transcription, rRNA processing, cellular factors, and the native cellular environment shape ribosome assembly and RNP assembly at large.

## 1. Introduction

Ribosomes are responsible for protein synthesis and are some of the largest and most complex macromolecular machines in a cell. Prokaryotic ribosomes are made up of a large subunit (LSU or 50S) and a small subunit (SSU or 30S). The *Escherichia coli* (*E. coli*) LSU consists of 23S and 5S ribosomal RNAs (rRNA) bound by 33 ribosomal proteins (r-proteins), while the SSU consists of 16S rRNA and 21 r-proteins [1]. The assembly of a ribosome is a very complex and multistep process that consumes about 40% of a cell’s energy [2]. Assembly is initiated with the transcription of a primary rRNA transcript containing ~4500 nucleotides. Transcription is assisted by the rRNA transcription antitermination complex (rrnTAC), which reduces transcription pausing and prevents early termination [3,4,5]. The primary transcript is co-transcriptionally processed by multiple specific RNases to form the three rRNA fragments (16S, 23S and 5S rRNAs) [6,7,8,9] that simultaneously fold into secondary and tertiary RNA structures [10,11,12]. Co-transcriptional rRNA folding follows the vectorial (5′ to 3′) direction and allows for the sequential binding of r-proteins [13,14,15,16,17,18]. Co-transcriptional rRNA processing, rRNA folding, and r-protein binding is accompanied by the introduction of base modifications, such as pseudouridinylations and methylations [19,20]. Furthermore, these processes are assisted by multiple assembly factors, such as GTPases, helicases, and maturation factors [1,21]. Remarkably, it takes only about 2 min for the cell to assemble a functional bacterial ribosome [22]. Consequently, the assembly intermediates of this process are short-lived and contribute to only ~2% of the total ribosome population [23], making them difficult to study.

Ribosome assembly, and RNP assembly in general, is very difficult to investigate. Apart from the complexity of the process and the low abundance of assembly intermediates, many of the biomolecular interactions that form during assembly are transient and dynamic in nature and therefore difficult to capture biochemically and structurally. Furthermore, the assembly processes are often very heterogeneous and consist of multiple parallel assembly pathways.

Ribosome assembly has been studied over many decades and, despite its complexity and technical limitations, various aspects of the process are well understood. There are several reviews that provide detailed overviews of various aspects of the assembly process [1,20,21,24,25,26,27,28,29,30]. Here, we aim to provide a methodological perspective on studying ribosome assembly and the assembly of other RNPs, such as the spliceosome, various mRNPs, and large non-coding RNPs. We summarise the various biochemical, structural, and biophysical methods employed over the years for studying different facets of the ribosome-assembly mechanism, with a focus on bacterial ribosome assembly. This review highlights the exciting parallel between the evolution of our understanding of ribosome assembly and the technological advancements that have led to the development of new methods (Figure 1). We start by discussing in vitro reconstitutions that employ a bottom-up approach using minimal components to understand the assembly process in a very controlled manner. Time-resolved mass spectrometry, RNA structural probing, and cryo–electron microscopy have provided information on the kinetics of assembly and have permitted the structural visualisation of the assembly process at high resolution. Single-molecule experiments have become instrumental in understanding how the different processes are functionally coupled with each other as they allow us to follow complicated, multistep processes in real-time. We conclude our review by discussing approaches that we think will be required in the future to understand how the ribosome and other complex protein–RNA machineries are assembled so fast and efficiently in vivo.

## 2. Biochemical Reconstitutions

### 2.1. In Vitro Reconstitutions

In the early days of studying ribosome assembly, it was evident that a ribosome is a very complex machinery composed of multiple r-proteins interacting with rRNA. In order to understand its assembly, the two subunits of the ribosome were studied separately. In vitro reconstitution/omission experiments were performed by mixing purified rRNA with different sets of r-proteins and then purifying the resulting assembly intermediates via the ultracentrifugation of sucrose gradients [32]. Initial attempts to reconstitute these subunits indicated that the 30S can be reconstituted in a single step [17], while several heating steps and various Mg^2+^ concentrations were required to reconstitute the 50S [33,34]. The reconstituted ribosomes were tested for their ability to read polyU templates [35,36], form peptide bonds [37], or bind tRNA [38], suggesting that these in vitro reconstitutions provide active ribosomes. Reconstitution experiments indicated that the binding of r-proteins occurred in a sequential order and allowed for the organisation of the ribosome assembly into assembly maps (Nomura map for 30S and Nierhaus map for 50S) containing the thermodynamic binding dependencies of the various r-proteins [16,17,18]. In vivo experiments using cold-sensitive mutant strains and strains lacking r-proteins validated the assembly maps derived from the in vitro reconstitution methods [39].

### 2.2. In Vivo Mimicry

While these reconstitution efforts were successful in describing the in vitro thermodynamic assembly pathway, assembly was much less efficient and required unphysiological heating steps and buffer conditions. Furthermore, the reconstituted ribosomes were not tested for their ability to translate a complete mRNA [40]. Importantly, these experimental conditions did not properly mimic the in vivo situation. Inside cells, the rRNA is efficiently transcribed and co-transcriptionally processed, modified, and bound by r-proteins simultaneously [41,42,43,44]. This entire process is assisted by multiple assembly factors. Developments in the field of cell-free systems spearheaded by the Jewett Lab have been used to reconstitute ribosomes with high activity in near-native assembly conditions. An integrative ribosome synthesis, assembly, and translation (iSAT) assay combines co-transcriptional ribosome assembly and the subsequent translation of mRNA via the assembled ribosome in a single reaction, with GFP as a readout for the successful assembly of an active ribosome (Figure 2A–C) [40]. The iSAT reaction consists of a plasmid containing the entire rRNA operon initiated by a T7 promoter sequence, T7 RNA polymerase, all r-proteins purified from native ribosomes (TP70), a second plasmid coding for the reporter mRNA sequence (GFP), and cell extract (S150) containing all the cellular factors required for ribosome assembly and translation (Figure 2A). The cell extract allows for the correct processing [45] and modification [46] of rRNA. Since all the key components required in ribosome assembly (as well as other components, such as assembly factors that assist ribosome assembly) are then present, the assembly of the ribosome is expected to proceed in a native way i.e., the processes of transcription, rRNA processing, r-protein binding, and base modifications are expected to occur simultaneously and assisted by assembly factors. While earlier iSAT reactions had translational efficiencies of 20% when compared to in vivo-purified ribosomes [45], their efficiency can be improved to 70% by adding crowding and reducing agents to the iSAT reactions [47]. iSAT reactions have been further extended to include the synthesis of individual r-proteins [48], yet the assay still needs to be further developed to allow all r-proteins to be synthesised in the same reaction. Of note, iSAT reactions work efficiently despite using T7 RNAP instead of the native *E. coli* RNAP. Better mimicking the in vivo situation, future adaptions of iSAT should include the native *E. coli* RNAP in order to also properly reproduce the native rRNA transcription speed and pausing behaviour, which is assisted by the rrnTAC.

Inspired by the lab-on-a-chip approach, the one-pot iSAT reaction assay has also been performed on a chip to reconstitute 30S subunits in near-native conditions (Figure 2D) [49]. Genes encoding for r-proteins and rRNA were immobilised on a chip surface as DNA brushes along with anti-HA antibodies (Figure 2D, right panel). One r-protein at a time was designed as a fusion protein with a HA tag, and the rRNA was modified to include a Broccoli aptamer. All genes were transcribed, and r-proteins were translated locally at the surface. The resultant increase in fluorescence signal from the broccoli aptamer on the regions of the chip coated with anti-HA antibodies indicated that the rRNA was bound by the HA-tagged r-protein and all upstream binding r-proteins according to the Nomura assembly map (Figure 2D bottom, right panel). Using this approach, they could recapitulate the r-protein binding dependencies (Nomura map) and their binding kinetics (Figure 2E). They were also able to monitor the late stages of 30S assembly, including the binding of the mature 30S to the 50S.

In summary, biochemical reconstitutions are a powerful tool for investigating the intricate details of a specific process using a minimalistic system. Recent ribosome reconstitutions that mimic native conditions have enabled researchers to study mutant ribosomes [50], incorporate non-canonical amino acids [51], and investigate the process of evolution in the context of ribosome assembly and function [52]. Furthermore, these methods can enable the investigation of the role of various assembly factors in wild-type versus mutant ribosomes and the engineering of new ribosomes with specific functions.

## 3. Mass Spectrometry

While the in vitro reconstitution/omission experiments allowed for the construction of ribosome assembly maps that summarise thermodynamic protein-binding dependencies, they do not contain any information on its protein-binding kinetics during assembly. By combining quantitative mass spectrometry (qMS) with pulse-chase experiments using stable isotope labelling, it became possible to complement the thermodynamic r-protein binding dependencies with r-protein binding rates.

### 3.1. In Vitro Mass Spectrometry for r-Proteins

Pulse-chase qMS (PC-qMS) allows for the tracking of binding rates for all r-proteins to the rRNA in a single experiment [15]. The rRNA is incubated with heavy isotope-labelled r-proteins for a specific amount of time, and then chased with an excess of light isotope-labelled r-proteins to complete the assembly (Figure 3A, left panel). The completely assembled subunits are then isolated on a sucrose gradient and the value of the heavy-to-total protein ratio for each protein is determined using mass spectrometry and plotted as a function of time (Figure 3A, centre panel). The resulting binding curves provide the average binding rates for the individual r-proteins (Figure 3A, centre and right panels). By repeating these experiments at different protein concentrations and temperatures, the authors demonstrated that (1) RNA folding and protein binding occur at similar rates, (2) the rate-limiting steps for different proteins is similar at low or high temperatures, and (3) the final steps of 30S synthesis are limited by many different transitions. Similar experiments were performed with a pre-folded 16S rRNA that was pre-bound with a subset of r-proteins [53]. They observed multiphase binding kinetics of r-proteins, suggesting further complexity in the assembly pathway. Their observations also indicate the presence of multiple assembly pathways and a delicate interplay between thermodynamic dependency and kinetic cooperativity. PC-qMS was also used to investigate the influence of assembly factors for the assembly of the 30S, showing, for example, that RimP allows for the faster binding of S9 and S19 but prevents the binding of S12 and S13, potentially by blocking their binding sites [54].

### 3.2. In Vivo Mass Spectrometry for r-Proteins and Assembly Factors

qMS-based methods were also applied to recapitulate the assembly pathway in vivo and for the identification of multiple assembly factors [22] (Figure 3B,C). The authors used an in vivo stable isotope pulse-labelling approach to characterise the exact r-protein composition of various populations of intermediates (Figure 3B). The cells were grown in heavy isotope media and pulse labelled with light isotope media. Various fractions from the sucrose gradient corresponding to assembly intermediates were digested by trypsin and subjected to qMS. The resultant in vivo data validated the presence of four assembly intermediates of 30S particles, as observed by Mulder et al. using in vitro reconstitutions. The 50S assembly was more continuous in cells and revealed six assembly intermediates, which indicated a general pathway where the 50S assembly starts opposite to the peptidyl transferase centre, forms intermediates where r-proteins are added globally to the whole structure, and ends with the formation of the central protrusion. Likewise, subjecting the fractions of a sucrose gradient to qMS analysis led to the identification of 15 known and 6 unknown assembly factors that co-occurred with specific assembly intermediates, indicating their role in that particular stage of assembly (Figure 3C).

MS was also used to understand the effects of cellular knockouts of assembly factors on the composition of ribosome-assembly intermediates. Experiments with strains lacking specific assembly factors showed a slower growth rate and an accumulation of assembly intermediates [54,55,56,57]. An investigation into in vivo assembly intermediates from mutant strains that lacked the assembly factors LepA or RsgA, for example, showed reduced levels of late-binding r-proteins, suggesting the role of these assembly factors in the late stages of assembly [58].

Apart from quantifying the composition of assembly intermediates, MS can also be used to investigate the post-translational modifications of r-proteins during assembly. For example, the Woodson Lab used MS to understand the extent of S5 and S18 acetylation during in vivo ribosome assembly and its effect on the formation of specific rRNA contacts [59].

qMS methods were also applied to study eukaryotic ribosome assembly. For example, Sailer et al. used multiple different affinity-tagged assembly factors to pull down and crosslink different intermediates of pre-60S particles from *Saccharomyces cerevisiae* [60]. A mass spectrometry analysis of the crosslinked peptides produced a protein–protein interaction map that identified the localisation of 22 unmapped assembly factors. The association based on relative abundances between the newly mapped assembly factors and specific intermediates indicated the approximate time at which they act in the assembly pathway.

### 3.3. In Vivo Mass Spectrometry for RNA Modifications

rRNA is modified by methylations as well as pseudouridinylations [19]. These modifications are deposited site-specifically by multiple different modification enzymes during the course of the assembly process. Traditionally, modifications are detected using reverse transcriptase primer extension techniques [61], or P1 nuclease digestion followed by thin-layer chromatography (TLC) or high-performance liquid chromatography (HPLC) [62,63]. Although these are very sensitive methods, they are tedious as they allow for the observation of only one modification at a time and are suitable for detecting only specific modifications. qMS analyses of RNA enable the detection of multiple site-specific modifications simultaneously. Typically, isotope-based labelling is used to detect the fraction of RNA molecules that is site-specifically methylated. However, the accurate quantification of lowly abundant modifications can be challenging. Furthermore, since pseudouridine is a structural isomer of uridine, it cannot be detected. Popova et al. used a metabolic labelling approach to validate methylations and detect pseudouridinylated residues [64]. CD_3_-methionine (the precursor to SAM) leads to a +3 Da mass shift that can be distinctly and confidently annotated. Similarly, 5,6-D-uracil leads to a −1 Da mass shift for a pseudouridinylated residue. Using this approach on assembly intermediates purified from cells, the authors were able to characterise the stages at which each residue is modified during the assembly process. For example, most of the modifications on the 23S rRNA occur early during assembly, as opposed to the 16S where the modifications are incorporated from a 5′ to 3′ direction, in agreement with a co-transcriptional rRNA modification process. Another study used qMS on S-adenosylmethionine (SAM; methyl donor used by methyltransferases)-depleted cells to study the importance of RNA modification on ribosome assembly [65].

Overall, mass spectrometry is a highly sensitive and quantitative method for determining the binding kinetics of r-proteins to rRNA as well as for studying when multiple r-protein or rRNA chemical modifications are introduced during assembly.

## 4. Electron Microscopy

Electron microscopy (EM) has been proven instrumental in providing high-resolution structural information on ribosome-assembly intermediates. Both for negative-stain and cryogenic EM, ribosome-assembly intermediates either from in vitro reconstitutions or purified from cells are applied to a grid for imaging. Optimally, the individual particles meant to be imaged are present in multiple different orientations to reconstruct a 3D image [66]. Seminal work by the Williamson Lab in 2010 demonstrated the potential of using structural information derived from a heterogenous population of assembly intermediates for understanding the mechanisms of ribosome assembly [67]. They performed time-resolved low-resolution negative-stain EM after mixing 16S rRNA with all 30S r-proteins and then freezing them at different time points. They were able to visualise 14 different assembly intermediates, which were classified into 4 major groups (Figure 4A). The population of the first group, representing the smallest assembly intermediate, decreased over time. The second group peaked at several minutes, while the third and fourth groups appeared only at later time points. In combination with PC-qMS, they were able to reconstruct a detailed assembly pathway for the 30S subunit in vitro, demonstrating multiple parallel assembly pathways (Figure 4A).

The resolution revolution in 2013 led to significant improvements in electron detection technology and reconstruction algorithms [70,71]. This enabled its use in investigating more heterogeneous populations of complexes present in the same sample, providing the basis for imaging multiple assembly intermediates that populate the 50S assembly pathway both in vitro and in vivo.

The in vitro reconstitution of 50S is a two-step process that leads to the activation of 50S [72]. High-resolution cryoEM of the two-step reconstitution process displayed five main classes (subpopulations) resulting from the first step and a mature 50S structure resulting from the second. 50S assembly initiates at its core, followed by the L1 protuberance and the central protuberance (CP). Interestingly, the main difference between the last class from step one and the fully mature 50S is a structural rearrangement of the rRNA that leads to the maturation of the peptidyl transferase centre. 

In order to perform experiments in native conditions and gain perspective on ribosome assembly in vivo, Davis et al. used high-resolution cryoEM of assembly intermediates isolated from a bL17 (r-protein of 50S)-depleted strain to enrich intermediates [73]. Sub-population averaging revealed that, similar to that of the in vitro experiments, the in vivo 50S assembly consists of a heterogeneous ensemble of intermediates. The different subpopulations that progressively evolved into a more mature complex could be further grouped together, thus providing structural evidence of parallel pathways of 50S assembly. Interestingly, a reanalysis of this compositionally and conformationally heterogenous data using a neural network-based framework called CryoDRGN revealed a previously unreported assembly intermediate [74]. CryoDRGN is a powerful tool that enables the automated classification of various states, which is typically performed using multiple manual and expert-guided rounds of hierarchical 3D classification.

A correlative analysis using qMS and cryoEM data from bL17 depleted cells also indicated that the unidentified densities in subpopulations from one of the assembly pathways corresponds to assembly factor YjgA [73]. Putative YjgA binding blocked the docking of a helix crucial for inter-subunit bridge formation, suggesting that YjgA acts as a late-stage assembly factor for maturation. Recent evidence suggests that the presence of assembly factors in vivo directly affects the order of maturation of specific regions. For example, in contrast to those of in vitro assembly [72], the core and the central protuberance formations were suggested to be interdependent in vivo [68,75]. Another detailed characterisation of pre-50S assembly intermediates revealed a network of assembly factors, such as ObgE, RsfS, YjgA, RldU, and YhbY, that orchestrate 50S maturation (Figure 4B) [68]. Several other studies have used cryo-EM to determine structures of bacterial ribosome-assembly intermediates to understand the function of assembly factors but are not further reviewed here [76,77,78,79,80,81,82,83].

Apart from structurally characterising later assembly intermediates that are formed once transcription is already completed and the majority of the rRNA is already processed, recent structural work has also provided information on the process of early rRNA transcription by the rRNA transcription antitermination complex and on the mechanism of initial rRNA processing. The rrnTAC is the macromolecular machinery responsible for the efficient transcription of rRNA in the cell [3,4,5]. The rrnTAC assembles on RNA polymerase (RNAP) and reduces NusA-mediated transcriptional pausing, R-loop formation, polymerase backtracking, and intrinsic as well as Rho-dependent termination. It enables chaperone-mediated rRNA folding and the formation of long-range rRNA–rRNA interactions. The high-resolution cryoEM structures of an in vitro-reconstituted rrnTAC-associated transcription complex revealed the presence of NusA, NusB, NusE, NusG, SuhB, and S4 (Figure 4C) [69]. Interestingly, in the rrnTAC, NusA is repositioned to prevent pausing caused by hairpin stabilisation as well as intrinsic termination. Similarly, the presence of NusG in the rrnTAC suppresses RNAP backtracking. The interactions of NusA, NusE, and SuhB with the C-terminus of NusG prevent it from recruiting Rho. Furthermore, the formation of a ringlike structure made by SuhB and S4 around the *E. coli* polymerase exit channel prevents Rho from directly interacting with the exit channel, therefore preventing Rho-dependent termination. Finally, the authors demonstrated that the 5′ end of RNA is bound by S4, and the emerging 3′ end of RNA is bound by Nus factors along with SuhB on the ring. This brings the distal regions of RNA close in space to form the long-range interactions that are required for creating a substrate for rRNA processing by RNase III [84].

The 4500-nucleotide-long primary rRNA transcript is initially processed by dsRNA-specific RNases to generate the pre-rRNA fragments that further mature into 16S, 23S, and 5S [1,8,9]. In *B. subtilis*, a mature 23S is obtained by Mini-III and a mature 5S by M5 processing [85,86]. The structural characterisation of these RNases with their respective substrates revealed that the Mini-III binds pre-23S ds-rRNA as a dimer, where each subunit cleaves one of the strands [87]. In contrast, the N-terminal domain of M5 binds to the 3′ strand of ds-rRNA and cleaves it. This then leads to structural rearrangements enabling the C-terminal domain of M5 to bind and cleave the 5′ strand. Mini-III and M5 are assisted by r-proteins, such as uL3 and uL18, that bind to the respective substrates and keep them in a conformation that can be recognised by the enzymes.

The above examples highlight the role of cryoEM as a powerful structural method for increasing our understanding of the structural and mechanistic details of ribosome assembly and for providing time-resolved information.

## 5. RNA Structure Probing

Multiple different studies have indicated the role of rRNA secondary and tertiary structures in the binding of r-proteins. The simple chemical or enzymatic probing of rRNA structures is a powerful method but has limited time-resolution and throughput [88]. The structural determination of rRNA during assembly is also difficult due to the presence of a heterogeneous set of conformations and the difficulty in resolving flexible regions. Hydroxyl radical footprinting (HRF) and DMS/SHAPE probing are two complementary methods that provide information on RNA tertiary and secondary structures, respectively. These methods overcome some of the above-mentioned limitations and have successfully been used to obtain more-detailed and better-resolved information on rRNA folding.

### 5.1. In Vitro RNA Structure Probing

RNA secondary structure information can be obtained using chemical reagents such as DMS and many other probes (e.g., glyoxal, 1m7, 1m6, NMIA, BzCN, NAI), which react specifically with single-stranded RNA but do not chemically modify dsRNA [89]. The introduced adducts result in a stop when read by a reverse transcriptase. The resulting fragments are detected using primer extension. This approach has been used to predict 16S rRNA secondary structures with 97% accuracy [90]. In order to increase throughput, an alternate strategy, termed SHAPE-MaP, uses manganese ions during the reverse transcription step, which causes reverse transcriptase to introduce a mutation into cDNA rather than stop at the modified sites [91]. The cDNA is sequenced and the percentage of the underlying mutations is used to generate a reactivity profile for predicting the secondary structure (Figure 5A). SHAPE-MaP was used to track the rRNA structure during ribosome assembly. For example, the SHAPE-MaP-based structural probing of 23S rRNA in the presence and absence of r-proteins showed very similar reactivity profiles, suggesting that the 23S rRNA assumes its secondary structure even in the absence of r-proteins [73].

While most of the chemical reagents introduced above require seconds to several minutes to react with their RNA substrate, and therefore limit the time resolution, hydroxyl radical footprinting (HRF) provides information at few-milliseconds resolution and therefore allows for the study of very early rRNA folding events and of the formation of protein–RNA interactions [92,93]. In HRF, rRNA is exposed to short pulses of hydroxyl radicals that are generated by X-rays. These hydroxyl radicals react with the unprotected RNA backbone and thereby cleave the RNA into smaller fragments. A site-specific primer extension is used to amplify these fragments. The probability of cleavage depends on solvent accessibility, and therefore reports on the RNA tertiary structure and/or its interaction with proteins. HRF experiments were performed in a time-resolved manner by mixing 16S rRNA with all r-proteins and exposing the reaction to an X-ray pulse at different time points after mixing, thus providing the first time point as early as 20 ms after mixing (Figure 5B–D) [92]. Apart from validating the kinetics of early and late r-protein binding as determined using PC-qMS, these experiments showed that 30S assembly nucleates from different points along rRNA (Figure 5C). Additionally, the authors observed that the initial encounter complexes refold during assembly. For example, S7 initially binds in a non-native conformation (protecting only H43 within 20–50 ms) and adapts a native conformation (protecting H29, H37, and H41) only much later in assembly (Figure 5D). These experiments demonstrate the potential for HRF to provide information on RNA structural changes at a nucleotide resolution and at a millisecond-to-second timescale.

rRNA folds into a very heterogeneous set of conformations during ribosome assembly [10,11,94,95]. Therefore, RNA structural probing will provide an average of all conformations present in the sample to be probed. While this has not yet been performed in studying rRNA folding during ribosome assembly, recent analysis pipelines have shown the potential for dissecting RNA heterogeneity using the property of DMS to achieve multi-hit kinetics and using single-molecule sequencing as a read-out [96,97,98,99,100].

**Figure 5 biomolecules-13-00866-f005:**
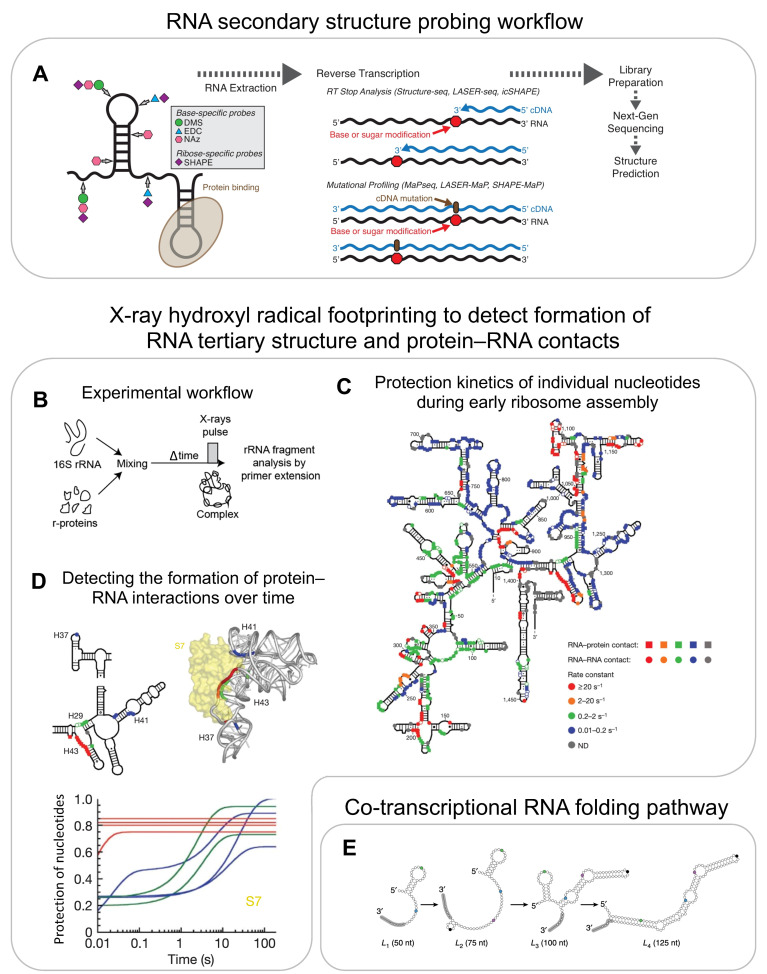
**RNA secondary structure determination using chemical probing and high-throughput sequencing:** (**A**) general workflow of RNA secondary structure probing. (**B**–**D**) **Tertiary structure and protein–RNA interaction determination using hydroxyl radical footprinting (HRF):** (**B**) experimental setup of in vitro time-resolved HRF. (**C**) Protection rates of individual residues of the 16S rRNA representing formation of RNA–RNA tertiary contacts as well as RNA–protein contacts. (**D**) Kinetics of rRNA backbone protection as a result of S7 binding represented on secondary (top left) and 3D (top right) structures and normalised fitted curves indicating the protection of residues (y-axis) as a function of time (x-axis) (bottom). Colour codes for Figure 5D are the same as indicated in Figure 5C. **Co-transcriptional RNA folding intermediates:** (**E**) model for co-transcriptional folding of the SRP RNA as determined by co-transcriptional SHAPE-seq. (**A**) is reproduced and adapted with permission from [101]. Reproduced with permission: (**C**,**D**) from [92], and (**E**) from [102].

### 5.2. Co-Transcriptional RNA Structure Probing

RNA probing assays have been performed on pre-transcribed rRNA, but rRNA folds co-transcriptionally in vivo and is affected by the speed of transcription [103]. While it has been shown that co-transcriptional rRNA folding is different from the folding of a pre-transcribed RNA [10,11,95], the structural probing of rRNA has not been performed in the context of transcription yet. However, co-transcriptional probing has been employed to study relatively simpler systems that undergo ligand-induced conformational changes, such as the fluoride riboswitch and the SRP RNA [102]. For this, DMS/SHAPE-based structural probing was adapted by designing roadblocks on the 3′ end of a DNA transcription template. The roadblocks prevent polymerase from transcribing further. The reactivity profiles of RNA fragments that were transcribed from different lengths of DNA templates (made by placing roadblocks at different positions) allowed the authors to mimic co-transcriptional RNA folding pathways, simulating, however, an infinitesimally slow transcription rate (Figure 5E). 

### 5.3. In Vivo RNA Structure Probing

RNA structural probing has also been performed in vivo to understand how rRNA folds in the native cellular environment and in the natural context of rRNA transcription. Soper et al. used HRF to study how assembly factors affect rRNA structure formation during assembly [59]. They compared protection resulting from mutant strains that lacked assembly factors, such as RbfA and RimM, to wild-type strains to determine the putative binding site of these assembly factors. Furthermore, closely analysing the assembly intermediates of mutant strains helped discover the role of these assembly factors in the assembly pathway. These assembly factors lead to global structural changes at late time points in assembly, binding to the 50S inter-subunit interface and thus acting as a checkpoint for quality control.

In order to obtain information on early co-transcriptional rRNA folding in vivo, a protocol was developed that uses the metabolic labelling of cells to separate newly transcribing rRNA intermediates from the total pool of rRNA [104]. Transcriptionally inactive cells (in nutrient-poor media) are labelled with 4-thiouridine (4sU) right before feeding with rich media. This allows for the isolation of nascent-transcribed rRNA, which can be probed by DMS or HRF. The results of the in vivo DMS probing of nascent 16S rRNA recapitulated the general vectorial folding pathway and specifically provided information on rRNA interactions at 30 seconds resolution. The results of the in vivo HRF probing of nascent rRNA are expected to provide exciting insights into rRNA tertiary structure formation at milliseconds resolution. Furthermore, this approach can potentially be extended by developing cell-compatible and faster acting probes to achieve millisecond-to-second resolutions for secondary structure probing [105].

## 6. Single-Molecule Methods

The so-far discussed ensemble biochemical, biophysical, and structural methods have led to the detailed characterisation of the mechanism of ribosome assembly, including the order and kinetics of r-protein binding, the dynamics of RNA structure formation, and the structural characterisation of assembly intermediates formed at various stages of assembly. However, these ensemble methods provide an average over the individual molecules. The need for averaging leads to the following major challenges: (1) the heterogeneity of the ribosome-assembly process cannot be sufficiently resolved, i.e., it is not possible to separately monitor the trajectories along the reaction coordinates of the individual assembly pathways; (2) it is not possible to dissect how different molecular processes, such as transcription progression, RNA folding, and protein binding, are functionally coupled to each other; and (3) dynamic structural changes may not be resolved.

Single-molecule methods instead allow for tracking the activity of individual molecules over long time scales at high temporal resolutions, thereby directly following multistep processes in real-time and dissecting the heterogeneity. To observe a single-molecule for minutes to hours, molecules of interest are immobilised on the chemically functionalised surface of a glass coverslip, typically using a biotin–streptavidin/neutravidin interaction (Figure 6A) [106]. Fluorescently labelling the molecule on the surface or the ligands that can bind to the surface-immobilised molecules allows for monitoring the conformational changes, binding events, and enzymatic activities of the molecules in real-time using total internal reflection fluorescence (TIRF) microscopy to reduce the fluorescence background. A fluorescent resonance energy transfer (FRET) can be used to directly measure the distance changes between a donor and one or several acceptor dyes [107] and thereby, for example, inform on conformational changes as they happen in real-time [108].

### 6.1. Multicolour Single-Molecule Fluorescence Microscopy

Some of the initial single-molecule experiments investigated the folding of the H20–H21–H22 three-way junction of 16S rRNA on S15 binding [109]. The three-way junction was immobilised using one of the helices, and the other two helices (H22 and H21) were labelled with a donor and acceptor dye, respectively. In the absence of S15, the three helices adapt a planar conformation that results in the limited transfer of energy from donor to acceptor (low FRET). However, in the presence of S15, the helices form a non-planar tertiary structure that brings the two dyes closer and leads to high FRET efficiency. Further, using a fast buffer-exchanging system, the authors titrated the levels of Mg^2+^ ions to determine that the three-way junction reacts instantaneously to Mg^2+^ ion levels.

One and a half decades later, more sophisticated multicolour experiments allowed for the visualisation of multiple processes at the same time, specifically the simultaneous tracking of rRNA folding and r-protein binding. Kim et al. investigated the binding of S4 (primary binding r-protein) to a 5-way junction (5WJ) in the 5′ domain of 16S rRNA (Figure 6A) [94]. They used a similar helix labelling system as described above for H3 and H16, and additionally labelled the r-protein S4 with another acceptor. Using this approach, they showed that S4 initially binds in a low FRET state (non-native conformation) and then later transitions into a high FRET state (native conformation) (Figure 6B,C). Performing similar experiments on the 5WJ indicated that helix H3 initially adopts a flipped conformation that recruits S4. This then enables H3 to dock onto S4 and assume a native conformation, suggesting that S4 guides rRNA folding (Figure 6D).

Interestingly, similar experiments applied to the initial binding of S15 to the central domain H20–H21–H22 junction showed that the binding of S15 leads directly into a high FRET state that does not change over time [95]. This suggests that, unlike that of S4, the S15 binding site immediately folds into its native conformation upon recruitment of the primary binding protein S15.

Further multicolour experiments on the 5′-domain system highlighted that r-proteins can efficiently change the rRNA folding landscape [110]. Monitoring the recruitment of S4, S20, and S16 showed that S16 can be stably recruited to a complex consisting of S4 and S20. The stable recruitment of S16 leads to conformational changes that enable H12 to interact with H3, which prevents H3 from flipping out and stabilising the native conformation. Overall, these experiments showed that r-protein binding changes the energy landscape such that only certain barriers can be crossed and, thus, limits the conformational search space.

### 6.2. Co-Transcriptional Single-Molecule Imaging

Ribosome assembly occurs co-transcriptionally, and, thus, the processes of rRNA folding and r-protein binding are linked to transcription [41,42,43,44,111]. Duss et al. developed a method to simultaneously monitor the process of transcription elongation and r-protein binding to a nascent rRNA directly emerging from the RNAP [11,95]. To this end, a stalled transcription elongation complex was formed that consists of a DNA template labelled with dyes at the 3′ end, native *E. coli* RNAP, and nascent rRNA (Figure 6E, left panel). This stalled complex was obtained by initiating transcription using only three out of the four NTPs on a sequence missing the fourth nucleotide. The stalled transcription complex was then immobilised to the imaging surface through the 5′ end of its nascent RNA using a complementary biotinylated probe. The experiment was initiated by the addition of all four NTPs. The progression of transcription brings the fluorescently labelled 3′ end of the DNA template closer to the surface, which leads to an exponential increase in signal intensity (Figure 6E, left panel) as a result of an exponential increase in excitation in the evanescent field generated by total internal reflection when moving closer to the surface. A plateau in fluorescence intensity during transcription termination demonstrated that RNAP can stall for a few seconds before dissociating from the DNA template, which is identified as a sudden intensity drop in its signal to zero [95]. The authors then monitored, using real-time transcription, the elongation of the 16S rRNA H20–H21–H23 three-way junction and, simultaneously, the binding kinetics of S15 to the nascent RNA (Figure 6E, centre panel). They found that S15 can only bind once the full-length three-way junction RNA has been transcribed. A detailed characterisation of the S15 binding events revealed three populations of nascent RNA molecules: (1) natively folded RNA molecules that stably bound S15 immediately upon transcription of the full-length three-way junction, (2) partially folded RNA molecules that bound S15 transiently, and (3) misfolded RNA molecules that did not bind S15 at all. They further showed that pre-transcribed RNA has distinct properties compared to co-transcriptionally folded RNA [95].

While this study indirectly reported on RNA folding using protein binding kinetics as a read-out, direct information on rRNA folding was missing. In a follow up study, the authors developed an approach that allows for the simultaneous tracking of (1) transcription elongation, (2) the co-transcriptional folding of nascent RNA, and (3) the binding of one or two proteins to nascent RNA (Figure 6E) [11]. Studying the 3′ domain of 16S rRNA showed that the primary binding r-protein S7 first engages transiently with nascent RNA before becoming stably incorporated, which happens upon binding of the secondary and tertiary binding proteins. Furthermore, the authors observed that the binding of S7 was more efficient on smaller constructs as opposed to the full-length 3′ domain, indicating a higher tendency of longer rRNA to misfold and thereby preventing r-protein binding. Four-colour experiments then showed that the binding of S7 directly depends on the formation of a long-range helix (H28), which forms more efficiently if less RNA needs to be transcribed before the 5′ and 3′ halves of this helix can meet to form the long-range helix (Figure 6F,G). This directly demonstrated that the formation of long-range RNA interactions are impeded by the 5′ to 3′ directional process of transcription [112]. Remarkably, rRNA folding efficiency increased in the presence of the 3′-domain binding r-proteins, indicating that r-proteins can chaperone rRNA folding and guide the energy landscape of ribosome assembly.

A similar study on the 5′ domain of 16S rRNA showed that the primary binding r-protein S4 binds transiently to the transcribing rRNA, whereas S4 could bind stably to pre-transcribed rRNA [10]. This suggests that structures formed early during transcription are not competent to stably recruit S4. They also found that the addition of secondary binding r-proteins led to longer-lived S4 binding events. These studies together suggest that r-protein binding-based rRNA remodelling is a general mechanism of ribosome assembly. 

Other approaches to track co-transcriptional RNA folding have also been developed but have not been applied yet to study co-transcriptional ribosome assembly. For example, forming an artificial transcription bubble, the authors of one study were able to introduce two different fluorescent labels site-specifically into nascent RNA [113] to study the co-transcriptional folding of a thiamine pyrophosphate (TPP) riboswitch. A FRET signal was used to study different conformational states of the aptamer assumed during transcription, in the presence and absence of the TPP ligand. In a similar approach, an azido UTP was site-specifically introduced into RNA and linked to a dye using copper-free click chemistry [114]. This approach revealed an inverse relationship between transcription speed and the metabolite-dependent folding of TPP riboswitch.

In another elegant study, a superhelicase was used to simulate and study the co-transcriptional folding of an RNA ribozyme [115]. First, a fully transcribed RNA, site-specifically labelled with two dyes, was hybridised with a complementary strand of DNA. This RNA–DNA hybrid was then immobilised to the surface for single-molecule imaging in the presence of the superhelicase. Transcription was mimicked using the addition of ATP, which triggered helicase activity to make the RNA single-stranded in the direction from the 5′ to 3′. They were able to investigate the RNA transitioning from a single-stranded state (low FRET) to a secondary folded (intermediate FRET) and a tertiary folded state (high FRET). Helicase activity can potentially be matched to transcription speed, but it still lacks the native transcriptional pausing that can directly influence RNA folding.

### 6.3. Optical Tweezers

While single-molecule fluorescence microscopy studies are powerful for tracking co-transcriptional RNA folding and the binding of proteins simultaneously at relatively high throughput, they lack the ability for tracking transcription elongation at single-nucleotide resolution. Optical tweezers, instead, can trap biomolecules—for example, transcription complexes between two beads—and allow for the observation of transcription progression [116] and RNAP pausing at single-nucleotide resolution [117]. Optical tweezers have been used to characterise real-time co-transcriptional RNA folding to understand the switching function of the adenine riboswitch and the resultant changes in RNA conformation upon ligand binding [118]. Optical tweezers also provide information on the forces exerted by biomolecules. For example, to understand how r-proteins stabilise rRNA structures, they mechanically unfolded and folded an irregular stem in domain II of 23S rRNA [119] in the presence and absence of r-protein L20. They found that L20 made the rRNA more resistant to mechanical unfolding by acting as a clamp around both strands of the rRNA stem.

Overall, single-molecule methods are very sensitive and provide direct and quantitative information. They inherently resolve biological heterogeneity and provide high temporal resolutions for tracking small and fast conformational changes of flexible regions that are averaged-out by conventional structural methods. Importantly, they provide information on how several different processes are functionally coupled with each other and how different assembly intermediates are placed along a reaction coordinate.

## 7. Integrative Methods

Multiple different biochemical, structural, and biophysical methods have been employed in studying the complex, multistep process of ribosome assembly. Yet, none of the methods can independently provide information on the entire process. Here, we highlight a few selected examples that integrate various methods.

In order to study the assembly mechanism of the bacterial 50S subunit in vivo, Davis et al. used a depleted bL17 strain to accumulate 50S assembly intermediates [73]. High-resolution cryoEM was used to determine the structures of 13 assembly intermediates. However, missing densities in the structures of these immature particles precludes the ability to obtain information on RNA structure and the associated proteins in these presumably dynamic regions. They used SHAPE-MaP-based chemical structural probing data to determine that, in these assembly intermediates, the 23S rRNA had a native secondary structure. Interestingly, sequencing reads also showed that some of the rRNA was not completely processed in the assembly intermediates. This is in agreement with previous reports that suggest final rRNA maturation occurs very late in assembly [1,120,121]. In order to provide information on the protein composition of the structural blocks that were missing in the cryoEM maps, they performed qMS and showed that the majority of r-proteins are already bound to these dynamic regions, and these blocks just need to be docked to the rest of the subunit to become a mature 50S subunit. Finally, one of the major drawbacks of structural methods is their inability to give direct information on function. In this case, to determine if assembly intermediates are capable of maturing into functional subunits, Davis et al. pulse-labelled bL17-depleted cells with heavy labelled media and simultaneously induced bL17 production. As expected, the peak in the sucrose gradient of the bL17-depleted assembly intermediates disappeared completely and the native 70S peak increased in intensity. This native 70S peak had heavy labelled bL17 incorporated, indicating that the addition of bL17 can rescue the intermediate and complete the maturation process.

In another study, Soper et al. used a combination of hydroxyl radical footprinting and qMS to understand the role of cellular factors in RNA folding and ribosome-assembly quality control [59]. Hydroxyl radical footprinting experiments showed how the assembly factor RimM reduces the misfolding of 16S head during transcription in vivo. Instead, qMS allowed them to confirm that, in absence of RimM/RbfA, some tertiary r-proteins are missing from the assembly intermediates. Further, they observed that the acetylation state of S18 directly correlates with the folding of rRNA and the formation of specific RNA–protein contacts during assembly.

A more recent study used native co-transcriptional in vitro reconstitutions in cell extract (iSAT) and characterised 50S assembly intermediates using time-resolved cryoEM and qMS to quantify both r-protein composition and the status of rRNA modifications during assembly [46]. The structures derived from the iSAT reaction were highly heterogenous. Thirteen structures were classified, spanning from one of the smallest known assembly intermediates detected to date (made of 600 nts and 3 r-proteins) to the latest stages of assembly with a nearly complete 50S subunit. Remarkably, studies that perform in vitro reconstitutions from purified components [72,122], co-transcriptional in vitro reconstitutions with cell extract [46], and characterising intermediates in vivo [73] show similar assembly intermediates, providing a general consensus on the mechanism of 50S assembly.

Overall, integrating multiple methods is very powerful and crucial for mechanistically understanding ribosome assembly and the assembly of other RNPs in detail.

## 8. Future Methods

The combination of different biochemical, biophysical, and structural approaches has allowed us to understand in great detail how the very complex process of ribosome assembly works at the molecular level. Moving forward, the major challenges to solve are (1) understanding how different processes in ribosome assembly are functionally coupled with each other and (2) visualising the structure and dynamics of ribosome assembly in the dense native cellular environment. In the following section, we will discuss emerging methods that we think will help to address these challenges.

### 8.1. Multicolour and Multiscale Single-Molecule Methods

Single-molecule methods are uniquely suited for understanding how different processes are functionally coupled with each other. The multicolour single-molecule fluorescence microscopy approaches discussed above demonstrate the potential to track multiple processes simultaneously. For example, they allow us to understand how transcription, RNA folding, and protein binding are directly interconnected [11]. Moving forward, more-complex in vitro reconstitutions that include more factors and processes will become accessible. Furthermore, experiments in cell extracts that contain all cellular factors will bridge the gap with in vivo experiments.

Apart from developing more-complex multicolour single-molecule fluorescence experiments, the future will also include combining single-molecule experiments with force experiments such as optical tweezers. For example, combining the two single-molecule modalities may allow for the tracking of transcription elongation and RNA folding at single-nucleotide resolution and, in addition, correlate the binding of one or two proteins to co-transcriptionally folding rRNA. In a recent study, the authors used force changes as a readout to monitor the individual codon translocation of ribosomes on mRNA or the unwinding of mRNA secondary structures by ribosomes, and simultaneously monitored the binding of fluorescently labelled elongation factor EF-G (Figure 7A) [123]. As a further extension of this technology, LUMICKS has extended the imaging part from single-colour to multicolour fluorescent microscopy [124]. However, despite its power in studying multiple processes simultaneously, this method lacks throughput. The optical tweezer technology can only study one complex at a time. To study very complex and heterogenous systems, such as ribosome assembly, efforts will be required to increase its throughput and automation, such as the commercial introduction of microfluids by LUMICKS.

Mass photometry imaging is another single-molecule method that uses interferometric scattering to determine the mass of individual molecules [125]. This, in combination with other methods, could be useful for studying the size distribution of assembly intermediates during different stages of assembly.

Recent advancements in direct RNA single-molecule nanopore sequencing may provide new opportunities for understanding how and when RNA modifications are introduced during ribosome assembly. In this technique, voltage is applied to a pore located in a membrane so the resulting ionic current can be detected [126]. When RNA passes through the pore, the detected current changes depending on which nucleotide is passing. Similarly, modified nucleotides also lead to a change in current that is specific to each RNA modification. In principle, this allows for the direct detection of all modifications present on a single molecule of RNA. The direct sequencing of 16S rRNA successfully detected the presence of m^7^G and pseudouridine at the population level [127]. Current advances in data analysis methods have allowed for the study of multiple other modifications, such as and not limited to m^6^A, m^5^C, m^1^G, m^6^2A, I, Nm, and 2′-OMe [128,129,130]. Recent developments have highlighted the potential of nanopore sequencing for detecting multiple RNA modifications on the same molecule at single transcript resolution [128,131,132]. This opens up an avenue to investigate if there is a specific order in which RNA modifications are introduced. The chemical probing of RNA followed by direct RNA nanopore sequencing was used to predict RNA secondary structures [133]. Combining base modification detection with chemical probing-based RNA structure determination could allow for an investigation on how RNA modification and RNA structure formation are functionally coupled in RNP assembly [134].

**Figure 7 biomolecules-13-00866-f007:**
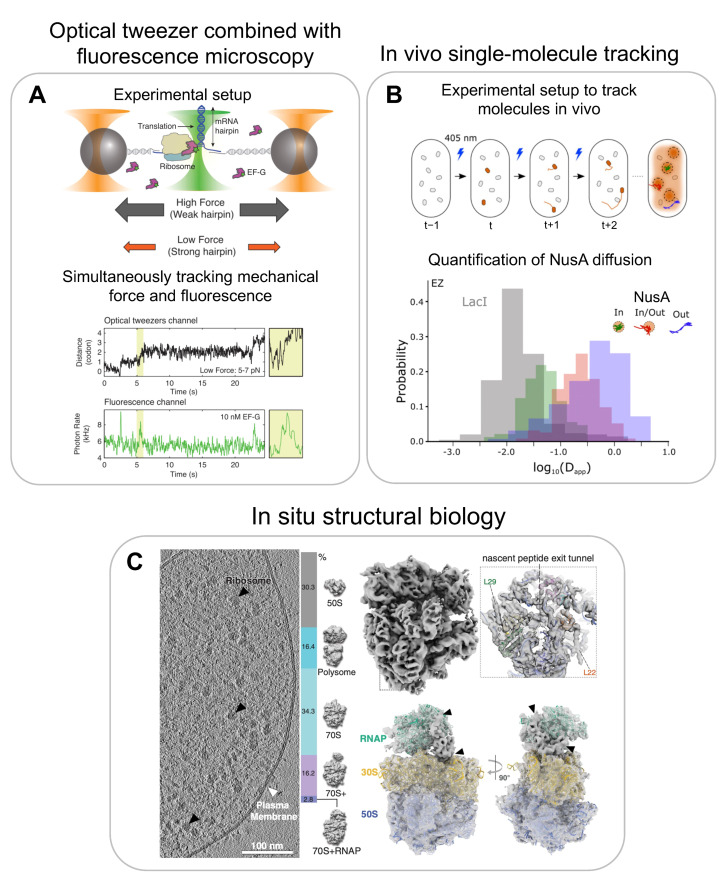
**Multiscale single-molecule methods for studying RNP dynamics at nucleotide resolution:** (**A**) experimental setup of optical tweezers combined with fluorescence microscopy for studying mRNA unwinding during translation (top); time traces indicating change in distance upon one codon translation (centre) and changes in fluorescence intensity upon elongation factor binding (bottom). **In vivo single-molecule tracking to study spatial localisation and dynamics:** (**B**) experimental setup of in vivo single-molecule tracking experiment (top); quantification of tracking data by plotting the distribution of the apparent diffusion coefficients, indicating dynamic movements of transcription factor NusA within and outside the presumable transcription condensates. **In situ structural biology:** (**C**) representative tomographic slice of an *M. pneumoniae* cell and quantitative classification of ribosome subtomograms (left); resultant structures of 70S (top right), and RNAP-ribosome supercomplex (bottom right). Adapted and reproduced with permission: (**A**) from [123] and (**C**) from [135]. (**B**) is adapted and reproduced from [136].

### 8.2. In Vivo Single-Molecule Tracking

For in vivo single-molecule tracking, individual molecules are not tethered to the coverslip, but molecules of interest are endogenously tagged with a fluorescent reporter and tracked in real time while they are moving within a cell [137]. The majority of molecules are much too abundant in the cell to be tracked all at once as a result of the diffraction limit of light. Therefore, a small subset of the molecules can be photoactivated first and then excited with a different wavelength for tracking. One common endogenous tag, which can be linked to the protein of interest, is mMaple3 [138]. This photoconvertible protein is activated by illuminating at 405 nm and can then be imaged by exciting the protein at 561 nm. Some initial studies looked at the clustering of RNA polymerase (RNAP) using in vivo single-molecule localisation to characterise the RNAP organisation inside cells. Interestingly, RNAP localisation experiments showed that the spatial clustering of RNAP is independent of rRNA transcription activity, as opposed to what was suggested earlier, but rather dependent on the underlying nucleoid structure [139]. Pushing this further, the transcription factor NusA, which is part of the rrnTAC involved in early ribosome assembly, was tracked in vivo [136] (Figure 7B, top panel). By evaluating the different single-molecule tracks and converting them to apparent diffusion coefficients, the authors found that NusA diffuses in three states: slow-moving molecules were assigned to the NusA molecules associated with the transcription complex, fast-moving molecules as freely diffusing, and a third class with intermediate mobility was assigned to the NusA molecules present in a transcription condensate, which likely forms by liquid-liquid phase separation. The individual components can freely diffuse in and out of these clusters, indicating that the droplets are dynamic (Figure 7B, bottom panel). These studies provided evidence that not only does eukaryotic ribosome assembly occur in a biomolecule condensate (nucleolus), but that a similar condensed state may also organise bacterial ribosome assembly. Such a mechanism could explain the much higher ribosome-assembly efficiency in vivo compared to in vitro reconstitutions.

Similar experiments were also applied for studying eukaryotic ribosome assembly (which occurs in both the nucleolus and cytoplasm [140]), for example, to track the export of pre-60S particles from the nucleolus to the cytoplasm through the nuclear pore complex [141]. The authors observed that transport is a single rate-limiting step and takes about 24 ms on average. Furthermore, the quantification of exports from single pores revealed that only one third of export attempts are successful, and the overall mass flux can be as high as 125 MDa per second.

Similar experiments could, in the future, allow us to track the dynamics of individual r-proteins or assembly factors to gain a better understanding of ribosome assembly in vivo. While single-molecule tracking can be extended to more than one colour, and recent break-throughs with the MINFLUX technology have maximised spatiotemporal resolution to nanometre spatial and submillisecond temporal resolutions [142,143,144], the requirement for the stochastic activation of single fluorophores in an ocean of otherwise unlabelled molecules makes it very unlikely that two differently labelled molecules would interact with each other. Therefore, directly tracking individual protein–RNA interactions or macromolecular conformational changes in vivo will require new technologies to be developed.

### 8.3. Cryo–Electron Tomography

Cryo–electron tomography (cryoET) is an emerging method for gaining structural understanding directly in native cellular contexts. CryoET uses the same basic idea as single-particle cryoEM to reconstruct 3D images. The main difference is that, in tomography, an image is acquired by tilting the sample at multiple different angles [145]. This provides images of the sample at multiple different orientations, which can be used to reconstruct a 3D image for each individual particle. This is in contrast with single-particle cryoEM, which typically uses averaged information from hundreds of thousands of particles present in different orientations [66]. Thus, cryoET can be used to look at individual complexes inside whole cells or sections of cells, thereby preserving their native structure.

For example, Xue et al. were able to identify *Mycoplasma pneumoniae* ribosomes during various stages of translation and provide a detailed map of the translation elongation cycle within a single cell [31]. Importantly, they were able to identify the specific translation state for each ribosome in the cell, providing spatial functional information on its translation status. They were able to quantitatively show that 26% of all ribosomes in their study were polysomes and determine the orientation of each ribosome in the polysome with respect to each other and their overall packing density. By comparing the individual ribosomes within a polysome, they could determine that the r-protein L9 of the leading ribosome adopts an extended conformation, protruding into the binding site of the translation elongation factors of the trailing ribosome and thereby providing a mechanism for preventing ribosome collisions. Applying similar approaches to the study of bacterial ribosome assembly in cellular contexts will be challenging due to the low abundance of ribosome-assembly intermediates compared to fully assembled ribosomes. Imaging cells treated with antibiotics to accumulate ribosome-assembly intermediates could be the first step to tackle this challenging problem. 

In another study from the Mahamid Lab, structures of an RNAP–ribosome supercomplex, termed expressome, were visualised in situ by combining cryoET with cross-linking mass spectrometry (Figure 7C, left panel) [135]. The structures showed for the first time how transcription–translation coupling is structurally organised in vivo (Figure 7C, right panel). They showed that the transcription factor NusA mediates coupling by physically linking the RNAP with the ribosome in *M. pneumoniae*. Furthermore, they visualised in high-resolution a state in which the ribosome has collided with the RNAP in the presence of an antibiotic that stalls the RNAP. Similar approaches could be used to visualise how bacterial ribosome assembly is coupled with transcription.

Eukaryotic ribosome assembly is separated from translation and takes place inside the nucleolus, which is a multiphasic biomolecular condensate that spatially organises maturing ribosome-assembly intermediates [140]. The Baumeister Lab used cryoET on native nucleoli of *Chlamydomonas reinhardtii* to show that pre-60S (LSU precursor) and SSU processome (SSU precursor) have different spatial localisation patterns. Furthermore, they classified three low-resolution structural assembly intermediates for each pre-60S and SSU processome. The maturation of these intermediates followed a gradient from the inside to the outside of the granular component [146].

Overall, these pioneering studies provide a starting point and demonstrate the potential for studying the complex process of ribosome assembly at high resolution in a native cellular context. Studying in vivo ribosome assembly could potentially answer questions such as the number of alternate pathways present in the assembly process and quantify the percentage flux in each of these pathways.

## 9. Conclusions

The assembly of a ribosome is a very complicated process involving the transcription, folding, modification, and processing of rRNA and the binding of dozens of r-proteins to nascent rRNA, assisted by dozens of assembly factors. Remarkably, the entire assembly process is completed within 2 min in the dense cellular environment. A plethora of biochemical, biophysical, and structural methods have helped further our understanding of this process in a quantitative manner: Sophisticated in vitro reconstitution systems in cell extracts that closely mimic the native process have been developed to bridge the gap between in vitro reconstitution from purified components and assembly in vivo. The use of pulse-chase quantitative mass spectrometry, time-resolved cryo–electron microscopy, and time-resolved RNA structure probing approaches has provided compositional and high-resolution structural data for understanding the kinetics of ribosome assembly and is instrumental in characterising multiple assembly intermediates along parallel assembly pathways. Recent multicolour single-molecule fluorescence experiments have shown the potential to follow how individual RNAs transcribe, simultaneously fold, and start to assemble into protein–RNA complexes in real time, providing information on how multiple different processes are functionally coupled with each other. Moving forward, in vivo single-molecule tracking, as well as cryo–electron tomography, will provide us with a much-needed understanding of how ribosomes assemble in their dense native cellular environment. Combining our efforts toward developing bottom-up reconstitutions of active systems that exhibit ever-increasing complexity with biophysical and structural approaches for visualising systems in vivo will bring us closer to understanding and, importantly, generating predictive models of how complex cellular processes work in a living cell [147].

## Figures and Tables

**Figure 1 biomolecules-13-00866-f001:**
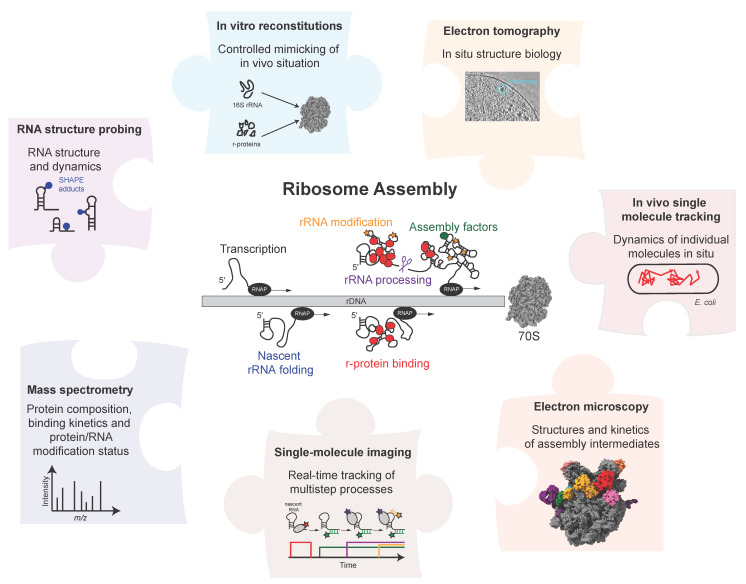
Overview of the biochemical, structural, and biophysical methods for studying ribosome and RNP assembly. Top right tomogram adapted and reproduced from [31], 70S (PDB: 4V6G) and 50S intermediate (PDB: 7BL5).

**Figure 2 biomolecules-13-00866-f002:**
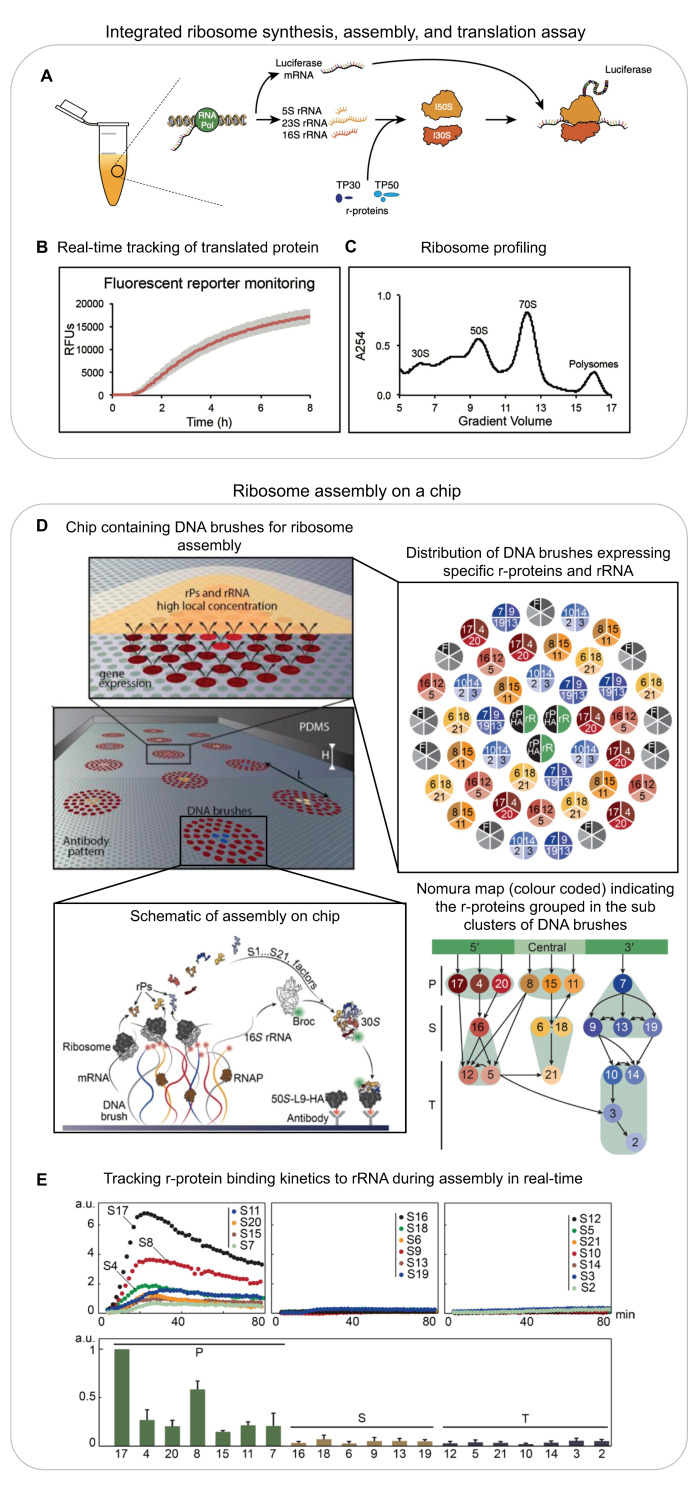
(**A**–**C**) **Integrated ribosome synthesis, assembly, and translation (iSAT):** (**A**) schematic of the one-pot iSAT reaction for the synthesis and assembly of ribosomes and the translation of a reporter protein; (**B**) real-time monitoring of fluorescence intensity as a reporter for translation activity of ribosomes produced in iSAT reaction; (**C**) ribosome profiling of the iSAT reaction. (**D**,**E**) **Ribosome assembly on a chip:** (**D**) schematic of chip surface and the distribution of DNA brushes (centre), zoomed-in schematic of one DNA cluster (top), distribution of DNA brushes (right) encoding for rRNA (black) and r-proteins-HA (green), assembly factors (grey) and other r-proteins colour-coded as in the Nomura map (bottom right), and schematic of ribosome assembly on chip (bottom centre); (**E**) time traces of primary, secondary, and tertiary r-proteins binding to rRNA during assembly on a chip (top: left to right) and normalised maximum signal from primary (green), secondary (yellow), and tertiary (grey) r-protein-HA (bottom: left to right). (**A**) is reproduced with permission from [40]. (**B**,**C**) are reproduced with minor adaptations with permission from [47]. (**D**,**E**) are reproduced from [49].

**Figure 3 biomolecules-13-00866-f003:**
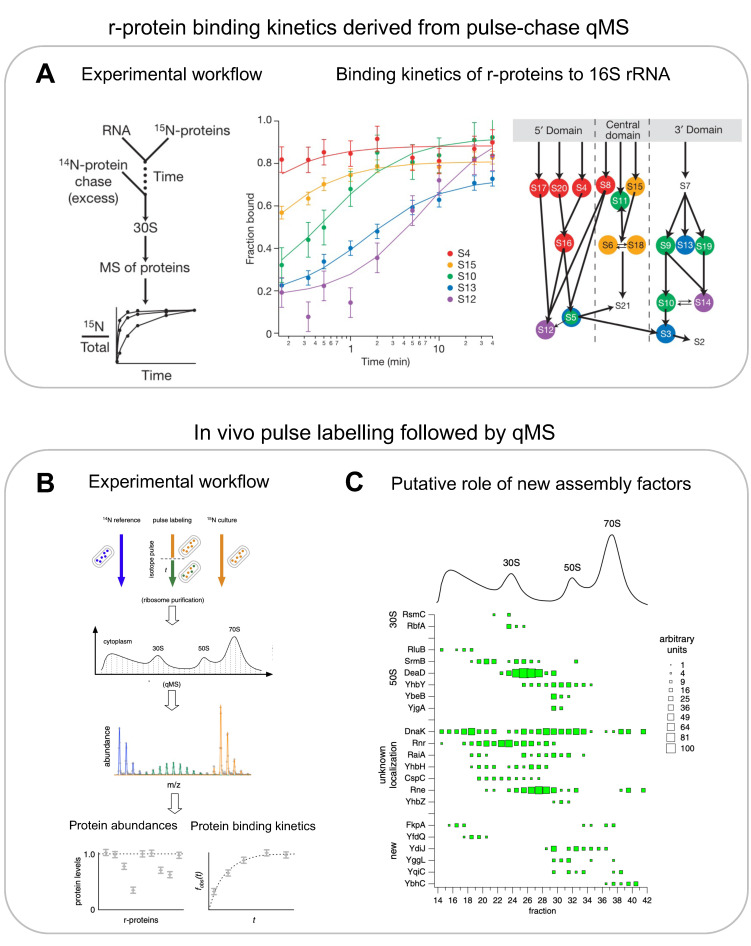
(**A**) **In vitro pulse-chase qMS to determine protein binding kinetics:** schematic of pulse-chase qMS workflow (left), r-protein binding curves to 16S rRNA (centre), Nomura assembly map coloured according to binding rates derived from pulse-chase qMS (right). (**B**,**C**) **In vivo pulse labelling to determine protein binding kinetics and discovery of new assembly factors bound to the ribosome-assembly intermediates:** (**B**) experimental workflow of in vivo pulse labelling and corresponding quantification by MS. (**C**) qMS-based identification and discovery of assembly factors and their potential role in assembly of specific subunits (right). (**A**) is reproduced with permission from [15], and (**B**) is adapted and reproduced and (**C**) is reproduced with permission from [22].

**Figure 4 biomolecules-13-00866-f004:**
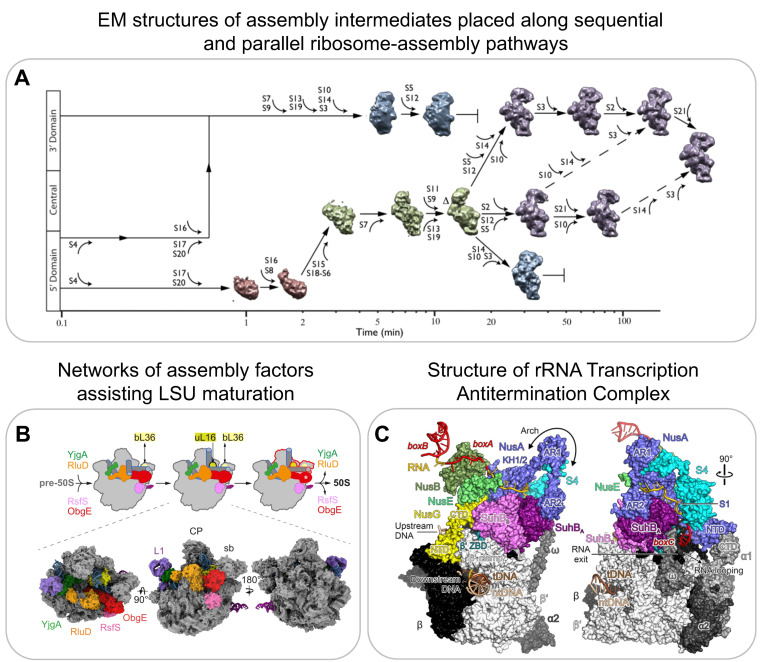
**Electron microscopy:** (**A**) Assembly intermediates populating parallel 30S assembly pathways visualised by negative-stain EM. (**B**) Model of how late-stage 50S maturation is guided by assembly factors (top); model can be constructed using several high-resolution cryoEM structures of pre-50S bound by assembly factors YjgA, RluD, RsfS and ObgE. (**C**) High-resolution cryoEM structures of the complete rRNA transcription antitermination complex (rrnTAC) responsible for efficient transcription of rRNA. Reproduced with permission: (**A**) from [67], (**B**) from [68], and (**C**) from [69].

**Figure 6 biomolecules-13-00866-f006:**
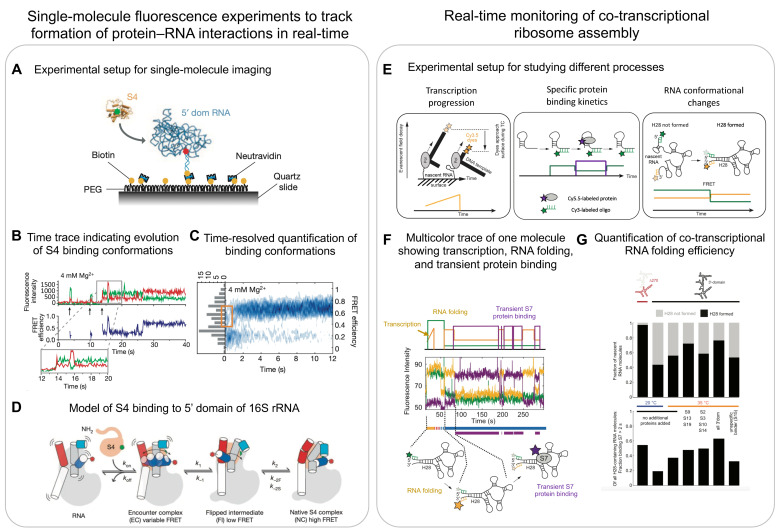
(**A**–**D**) **Single molecule fluorescence microscopy experiments for tracking changes to protein–RNA interactions in real-time:** (**A**) experimental setup of typical single-molecule experiments: schematic shows specific binding of S4 to the 5′ domain of 16S rRNA using single-molecule FRET. S4 was labelled with a donor dye (Cy3 in green) and the immobilised RNA by an acceptor dye (Cy5 in red). (**B**) Single-molecule trace of S4 binding to the 5′ domain of the 16S rRNA, leading to anti-correlated changes to the Cy3 and Cy5 channels over time. (**C**) Ensemble FRET efficiency plot highlighting a non-native intermediate state of S4 binding (orange box). (**D**) Proposed model of rRNA rearrangements upon S4 binding (bottom panel). (**E**–**G**) **Real-time tracking of multiple processes occurring during co-transcriptional ribosome assembly:** (**E**) experimental setups for simultaneously detecting transcription progression (left), specific protein binding kinetics (centre), and RNA conformational changes (right). (**F**) Multicolour single-molecule trace showing real-time transcription progression, long-range rRNA helix-28 (H28) formation, and transient binding of r-protein S7. (**G**) Quantification of single-molecule data from experiments shown in (**E**,**F**) under different conditions: the plots show the efficiency of H28 formation (top) and the efficiency of S7 binding to the subset of molecules that have H28 formed (bottom). (**A**–**D**) are reproduced with permission from [94] and (**E**–**G**) are reproduced and (**F**) is adapted with permission from [11].

## Data Availability

Not applicable.

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
