# Peer review of "Emerging Quantitative Biochemical, Structural, and Biophysical Methods for Studying Ribosome and Protein–RNA Complex Assembly"

_biomolecules, 2023, doi:10.3390/biom13050866_

Round 1

Reviewer 1 Report

Evaluation of the review article “Emerging quantitative biochemical, structural and biophysical methods to study ribosome and protein-RNA complex assembly” by Kavan Gor and Olivier Duss.

The author sum up in a comprehensive manner the various in vitro approaches to analyze ribosome assembly and protein-RNA interaction. The figures are of very good quality and the paper is well written and will be well received by the community. One minor criticism is that the paper is solely focused on bacterial (Escherichia coli) assembly and largely ignores data from other organisms, which is ok as long as this is stated in the title. The authors should therefore modify their title accordingly. Otherwise, I would miss some papers from the eukaryotic assembly field that should be included into the cryo-EM tomography section (e.g. Erdmann et al., 2021; Nat. commun.), the MS section (e.g. Sailer et al., 2022; Cell Rep.)  or the single-molecule tracking section (e.g. Ruland et al., 2021; Nat. commun).

Minor issues:

204: …cell strains…: delete cell

316: Interestingly, the….: delete Interestingly (repetitive usage of word)

320: Finally, they see that the 5’ end of RNA….: Finally, the authors demonstrated

466: ….system, they titrated the….system, the authors titrated

471: ….to a 5WJ in the…: to a 5-way junction (5WJ)

476: …5WJ junction indicated…..: 5WJ indicated

534: In a follow up study, they……: In a follow up study the authors…..

619: …they pulse-labelled…: refers to three references [1, 119, 120] from

 three different labs. Please specify.

775: please check that sentence

see above comments

Author Response

We thank the reviewer for their helpful input. Following their suggestion, we have now also added a couple of sections highlighting examples of eukaryotic ribosome assembly to make it a bit broader, for example, we now mention Erdmann et al. for cryoET, Sailer et al. for MS and Ruland et al. for single-molecule tracking. Nevertheless, we have now also clearly stated in the abstract and introduction that this review focuses mainly on bacterial ribosome assembly but can be used in future also for the assembly of other RNPs (we were asked by reviewer 4 to more stress the fact that it can be applied for studying other RNPs). We have also addressed all the minor requested corrections.

Reviewer 2 Report

Review “Emerging quantitative biochemical, structural and biophysical methods to study ribosome and protein-RNA complex assembly” by Gor and Duss is an up-to-date compilation of modern techniques applied to the problem of bacterial ribosome assembly. Authors themselves made a substantial contribution to the field. The expertise of authors in this topic is undisputed.

The selection of results to review is a matter of author’s choice. The preference was made for modern contributions, which make sense.

The only matter that screams for corrections is a matter of illustrative materials. Figures from the original experimental papers have been provided. Illustrations lack any attempts of unification. Majority of panels lack proper descriptions and distract a reader. E.g. Figure 2B: “Schematic of DNA brushes encoding for rRNA (black) and r-proteins-HA (green), assembly factors (gray) and other r-proteins (color-coded as in Figure 2C).” The panel shows a bunch of squares divided by sectors and labeled. I doubt a reader not familiar with the original work could gain any understanding of ribosome assembly while looking at this panel. This is only one example, but actually the MAJORITY of illustrations could not be understood by a reader not familiar with particular work.

Author Response

We thank the reviewer for their helpful input. We have now better organized all the figures and have added several additional explanations into the figure panels. For Figure 2B, we have also added additional panels for clarity. We hope that our significant modifications in the figures make them clearer.

Reviewer 3 Report

The manuscript is a beautiful read that covers a lot of ground and very well summarizes emerging methodologies enabling to better analyze Ribosome and RNP assembly. The authors have been doing a great job with the writing and illustrations. 

The manuscript is particularly centered on bacterial ribosome/RNP assembly. The author could probably make this clearer in the title/summary/intro.

Author Response

We thank the reviewer for their helpful input and are happy that they liked our writing and illustrations. To make it a bit broader, we have now also added a couple of sections highlighting examples of eukaryotic ribosome assembly. Nevertheless, we have now also clearly stated in the abstract and introduction that this review focuses mainly on bacterial ribosome assembly but can be used in future also for the assembly of other RNPs (we were asked by reviewer 4 to more stress the fact that it can be applied for studying other RNPs).

Reviewer 4 Report

This manuscript reviews the experimental methods that have been used to study bacterial ribosome assembly and highlights some of the major findings from these experiments.  The authors cover a range of studies from the classic assembly mapping experiments by Nomura and Nierhaus to recent developments in cryo-ET.  I liked this broad perspective, and I appreciate that older studies were included as well as very recent ones.  The emphasis is on experimental approaches rather than on concepts, but I think this is appropriate because the concepts behind cooperative, hierarchical assembly have already been reviewed a number of times.  

Overall, this review of methods for studying ribosome assembly is very informative and well organized. The text is clearly written and supported with nicely prepared illustrations. I think it will make a great addition to the literature and to this special issue on ribosomal proteins and ribosome biogenesis.

The manuscript does not need revision.  The authors, however, may wish to consider one editorial suggestion before publication:  They note in the abstract that bacterial ribosomes have been a proving ground for methods that are generally useful for studying the assembly of RNA-protein complexes.  I completely agree.  I think this could be emphasized more strongly in the introduction to the main article, perhaps around line 55 or line 70.  In addition to noting the relevance of these methods to other RNPs, they could also add one or two sentences highlighting the main challenges in dissecting RNP assembly pathways. Beyond noting the complexity of ribosome biogenesis, a sentence explaining why this complexity makes it hard to study would help those new to the field.

Line 326, B. Subtilis is mis-capitalized.

Author Response

We thank the reviewer for their helpful input and are happy that they find it very informative and well organized and appreciate that they find that the text is clearly written and supported with nicely prepared illustrations. Following their suggestion, we have now more strongly emphasized in the introduction that these methods are generally useful to study assembly of other RNPs. We also have added an additional paragraph highlighting the main challenges in dissecting RNP assembly. We have also corrected the typo.

Reviewer 5 Report

This well-written manuscript illustrates biochemical, structural and biophysical methods useful to investigate the molecular mechanisms of how ribosomes as well as protein-RNA complexes assemble. In my opinion, this is a very elegant review. Several methods are illustrated in a complete and clear manner. This manuscript appears useful in the field of ribosome investigations. 

Author Response

We are happy that the reviewer finds this a very elegant review with several methods illustrated in a complete and clear manner and no revisions were requested.